# Chronic Pulmonary Aspergillosis after Surgical Treatment for Non-Small Cell Lung Cancer—An Analysis of Risk Factors and Clinical Outcomes

**DOI:** 10.3390/jof10050335

**Published:** 2024-05-06

**Authors:** George Whittaker, Marcus Taylor, Mathilde Chamula, Felice Granato, Haval Balata, Chris Kosmidis

**Affiliations:** 1Department of Thoracic Surgery, Wythenshawe Hospital, Manchester University NHS Foundation Trust, Manchester M23 9LT, UK; marcus.taylor1@nhs.net (M.T.); felice.granato@mft.nhs.uk (F.G.); 2Division of Cardiovascular Sciences, University of Manchester, Manchester M13 9PL, UK; 3National Aspergillosis Centre, Department of Infectious Diseases, Wythenshawe Hospital, Manchester University NHS Foundation Trust, Manchester M23 9LT, UK; mathilde.chamula@nhs.net (M.C.); chris.kosmidis@nhs.net (C.K.); 4Manchester Thoracic Oncology Centre, Wythenshawe Hospital, Manchester University NHS Foundation Trust, Manchester M23 9LT, UK; haval.balata@mft.nhs.uk; 5Division of Immunology, Immunity to Infection and Respiratory Medicine, University of Manchester, Manchester M13 9PL, UK; 6Division of Evolution, Infection and Genomics, Faculty of Biology, Medicine and Health, Manchester Academic Health Science Centre, University of Manchester, Manchester M13 9PL, UK

**Keywords:** chronic lung infection, lung cancer, thoracic surgery, 1-year mortality, survival

## Abstract

Chronic pulmonary aspergillosis (CPA) is a rare but significant complication of lung cancer surgery. Its effect on survival remains unclear. Our aim was to describe the outcomes of the patients who developed CPA following the surgery for non-small cell lung cancer (NSCLC), identify the risk factors associated with its development following lung resection, and evaluate its impact on survival. All the patients with a diagnosis of CPA and operated NSCLC were identified in the National Aspergillosis Centre (NAC) database (2009–2020). Additional patients were identified in the Northwest Clinical Outcomes Research Registry (2012–2019) database. A regression analysis was performed to examine potential links between CPA and long-term outcomes and also to identify the factors associated with the development of CPA. The primary outcomes were the development of CPA, 1-year and 5-year mortality, and overall survival. Thirty-two patients diagnosed with CPA after lung resection were identified in the NAC database, of which 11 were also contained within the NCORR database, with a prevalence of 0.2% (*n* = 11/4425). Post-operative CPA was associated with significantly lower survival on log-rank analysis (*p* = 0.020). Mortality at one year was 25.0% (*n* = 8) and 59.4% (*n* = 19) at five years after the CPA diagnosis. On univariable analysis, a lower mean percentage-predicted forced expiratory volume in 1 s, ischaemic heart disease, and chronic obstructive pulmonary disease were all significantly associated with CPA development. CPA is a rare complication following lung cancer surgery which has a significant impact on long-term survival. Its development may be associated with pre-existing cardiopulmonary comorbidities. Further research in larger cohorts is required to substantiate these findings.

## 1. Introduction

Lung cancer is one of the most common cancers worldwide, accounting for 22% of all cancers in the United Kingdom, and is the biggest cause of cancer mortality [1]. The recommended treatment for early-stage non-small cell lung cancer (NSCLC) is surgical resection, when feasible, which carries a number of morbidity risks including infectious complications that can negatively impact long-term survival [2].

Chronic pulmonary aspergillosis (CPA) is a slowly progressive, destructive fungal infection of the lung parenchyma that typically develops in patients with pre-existing lung diseases such as chronic obstructive pulmonary disease (COPD) and interstitial lung disease (ILD) [3]. Previous studies have demonstrated that lung cancer surgery is also an independent risk factor for the development of CPA [4,5]. However, it remains unclear whether the development of CPA in this context affects long-term survival [6]. A number of risk factors have been identified for the development of CPA following lung resection, including active smoking, underlying respiratory comorbidities, and peri-operative respiratory complications [4]. The clinical presentation, disease burden, and long-term outcomes of the patients who develop CPA following lung cancer surgery are not well described in the literature, perhaps due to its limited incidence. The objectives of this study were to describe the clinical characteristics and outcomes of the patients who developed CPA following the surgery for NSCLC, identify the risk factors associated with its development, and examine the impact of CPA on overall survival after lung resection.

## 2. Materials and Methods

All patients referred to the National Aspergillosis Centre (NAC), Manchester, UK, between January 2009 and March 2020, who were subsequently diagnosed with CPA and had a prior history of surgery for NSCLC, were identified from the clinical database and included in the analysis. CPA was diagnosed at the time of the first clinic visit at the NAC for the patients who fulfilled all three of the following diagnostic criteria set by international clinical guidelines [7]: (1) progressive cavitary changes on a chest CT scan; (2) a positive sputum culture or positive *Aspergillus* IgG; and (3) the exclusion of alternative diagnoses. All patients with new radiological findings had CT scans reviewed by a chest radiologist and the cases were discussed at the local respiratory and lung cancer multidisciplinary team meetings. Additionally, all consecutive patients who underwent lung resection for primary NSCLC (confirmed on post-operative histology) at Manchester University NHS Foundation Trust between January 2012 and December 2019 were identified in the Northwest Clinical Outcomes Research Registry (NCORR) database and included in the analysis. These patients from the two datasets were cross-referenced to identify those with a recorded diagnosis of CPA and lung cancer surgery. 

The study received ethical approval from the University of Manchester (2021-10998-18442, 21 March 2021) and was approved by the NCORR steering committee. The NCORR database has full ethical approval from the regional Research Ethics Committee of the Health Research Authority. Individual patient consent was not required due to the retrospective nature of the study and the anonymised fashion of the data.

The primary outcomes were the development of CPA following surgery, 1-year mortality, 5-year mortality, and overall survival. 

The variables with more than 20% of data missing were excluded. Missing categorical data were imputed using an approach whereby it was assumed that missingness was equal to absence. Missing continuous data were handled using multiple imputation in accordance with Rubin’s rules [8]. The analysis of the difference in overall survival between the CPA and non-CPA groups, as well as within the CPA group, was assessed using the log-rank analysis and a survival curve was generated using the Kaplan–Meier method. The limited number of events in the CPA group precluded multivariable analysis from being undertaken. However, univariable analysis using logistic regression was performed to identify factors significantly associated with the development of CPA, and factors significantly associated with mortality following CPA diagnosis within the CPA cohort. Adjusted odds ratios (OR), hazard ratios (HR), and 95% confidence intervals (CI) were calculated, respectively. All tests were two-sided and statistical significance was defined as *p* value < 0.05. All statistical analysis was undertaken using SPSS version 28 (SPSS, Inc., Chicago, IL, USA).

## 3. Results

### 3.1. Patient Characteristics

A total of 32 patients with a subsequent diagnosis of CPA following surgery for lung cancer were identified from the NAC database. In terms of respiratory comorbidities, 84.4% (n = 27) had a history of COPD, whilst none had a history of tuberculosis or ILD. With regard to lung cancer histology for which the patients received surgery, 46.9% (n = 15) had adenocarcinoma, 34.4% (n = 11) had squamous cell carcinoma, and 18.8% (n = 6) had an alternative subtype. 

Overall, 56.3% underwent a right-sided resection (n = 18) and 74.2% (n = 23/32) underwent an upper lobe resection. One patient underwent a left-sided pneumonectomy, one patient underwent a right-sided upper bilobectomy and one patient underwent a non-anatomical wedge resection.

### 3.2. CPA Diagnosis

The median time from cancer surgery to CPA diagnosis was 4.6 years and was not significantly different based on histology (4.1 years for patients with adenocarcinoma and 4.5 years for patients with squamous cell carcinoma). For those who had adjuvant chemotherapy, the median time to CPA diagnosis was 3.2 years, which was significantly shorter than the 5.2 years for those who did not (*p* = 0.021). Finally, for patients who underwent radiotherapy, the median time to CPA diagnosis was 4 years compared to 4.9 years for those who did not (*p* = 0.457).

Upon CPA diagnosis, 46.9% (n = 15) had an aspergilloma on CT imaging. A total of 84.4% (n = 27) presented with dyspnoea, 71.9% (n = 23) presented with a productive cough, 53.1% (n = 17) had weight loss, and 28.1% (n = 9) had haemoptysis. CPA developed in the ipsilateral hemithorax in 87.5% (n = 28) of cases and in the contralateral hemithorax in the remaining 12.5% (n = 4) of cases.

Among the 62.5% (n = 20) of patients who had a fungal culture sent at diagnosis, 35% (n = 7/20) had a growth of azole-sensitive *Aspergillus fumigatus*. The median *Aspergillus* Immunoglobulin G (IgG) on diagnosis was 154 mg/L (range 28–987) (normal range < 40 mg/L ImmunoCAP, ThermoFisher Scientific, Waltham, United States).

### 3.3. Outcomes

All patients were treated were oral azole antifungals as per local guidelines. Twelve (37.5%) patients completed their course and remained stable with no sign of CPA relapse on follow-up after a mean of 17 months of treatment. Seven (21.9%) patients were stable but were on ongoing antifungal treatment for a mean of 30 months. Four (12.5%) patients had clinical progression on treatment, 12.5% (n = 4) of the patients died before assessment of CPA response could be made, and 6.3% (n = 2) of the patients were diagnosed with cancer recurrence. Three (9.3%) patients were lost to follow-up. 

Mortality at one year was 25.0% (n = 8) and 59.4% (n = 19) at five years after CPA diagnosis. The five-year outcome data were available for all 32 patients. On univariable analysis, a prior diagnosis of squamous cell carcinoma was associated with higher mortality (OR 3.1, 95% CI 1.2–7.6, *p* = 0.015). Male gender, age greater than 65, COPD, diabetes, and *Aspergillus* IgG > 200 mg/L were not significantly associated with overall survival.

### 3.4. Patient Characteristics within the NCORR Database

From the 32 patients described in the NAC database above, cross-referencing revealed that 11 were also enrolled in the NCORR database, which comprised a total of 4425 patients. The mean age was 66.7 years (±10.8 years) and 48.2% (n = 2135) were male. The prevalence of CPA within the NCORR database was 0.2% (n = 11) and the median time from surgery to CPA diagnosis was 23 months (range 9–46 months). 

Patients with CPA had a significantly lower incidence of any previous non-lung malignancy diagnosed or treated prior to their lung cancer diagnosis. There was also a significantly lower mean percentage-predicted forced expiratory volume in 1 s (FEV1) and a significantly higher incidence of ischaemic heart disease, COPD, and peripheral vascular disease. Complete patient characteristics are shown in Table 1.

The median follow-up period was 29 months (IQR 15–55 months). The estimated median overall survival for the CPA cohort was 47 months (95% CI 26–68 months). Mortality at 1 year was 0% and at 5 years was 57%. The estimated median overall survival for the non-CPA cohort could not be calculated as more than 50% of the patients were alive at the time of writing. On univariable analysis, the presence of post-operative CPA was associated with significantly reduced overall survival (log-rank analysis, *p* = 0.020), as shown in Figure 1. This effect was confirmed on Cox univariable analysis (HR 2.353, 95% CI 1.119–4.949, *p* = 0.024).

Given the limited sample size of the CPA group (n = 11) multivariable analysis was not appropriate. However, univariable analysis using logistic regression was undertaken to identify the factors potentially associated with the development of CPA. These results demonstrate that a lower mean percentage-predicted FEV1 (*p* = 0.010), ischaemic heart disease (*p* = 0.033), and COPD (*p* = 0.015) were all associated with a significantly higher risk of developing CPA. A lower percentage-predicted diffusion capacity of the lung for carbon monoxide approached, but did not reach, statistical significance. These results are displayed in Table 2.

## 4. Discussion

This study has demonstrated that CPA is a rare complication of lung resection for NSCLC but is associated with reduced long-term survival on univariable analysis. However, the small patient numbers precluded multivariable analysis and hence this association has not considered the presence of other risk factors. Nevertheless, these results do suggest that the development of CPA is potentially associated with an adverse prognosis after lung resection for NSCLC and hence further study in this area is warranted.

These findings are in keeping with the previously published literature, which demonstrated a significant reduction in overall survival on univariable analysis in 93 patients who developed CPA following lung resection amongst a cohort of 6777 patients. However, this effect was not retained when a multivariable analysis was undertaken [6]. CPA developing as a sequela of surgery for lung cancer appears to have a worse prognosis than CPA linked to other causes. In our cohort, a mortality of 25% after one year of CPA diagnosis was observed. Another study also reported poor one-year survival rates: 53% mortality in a cohort of 17 patients [5]. These findings may be attributable to the low treatment response observed in the aforementioned study, with only 35% of the patients demonstrating a positive response to antifungal treatment. Conversely, in an unselected cohort of CPA patients, mortality was only 7% at one year [8]. In our cohort, approximately 60% responded to CPA treatment, which is comparable to what is reported in the literature for CPA [9]. However, a proportion of the patients (12.5%) died before an assessment of the response to the CPA treatment could be undertaken. 

In this study, CPA was diagnosed a mean of 4.6 years following lung cancer surgery, which was longer than that reported in another study, although identifying the exact onset and duration of symptoms can be challenging due to underlying respiratory disease [4]. This did not differ significantly between cancer types but was significantly shorter for patients who had adjuvant chemotherapy, which is likely related to its immunosuppressive effects. As expected, patients with squamous cell carcinoma had a higher mortality compared with adenocarcinoma, whereas *Aspergillus* IgG and respiratory comorbidities did not have an effect on outcomes.

A worse lung function and the presence of cardiorespiratory comorbidities have emerged as potential risk factors for the development of CPA in this cohort of patients. The presence of COPD was also found to be associated with the development of post-operative CPA in other studies [5,6], suggesting that emphysematous lung parenchyma increases the risk of developing cavitary lesions. This is unsurprising given the well-established link between CPA and pre-existing lung diseases such as COPD [10], and may explain the poor survival rates. Indeed, Tamura et al. [5] found the presence of ILD to be the only other risk factor (in addition to COPD) associated with CPA development after multivariable analysis: we were not able to examine this in our study as no patient in this cohort had a diagnosis of ILD [5]. Conversely, Shin et al. identified several risk factors on multivariable analysis. These included low body mass index, smoking, ILD, undergoing surgery via thoracotomy, early post-operative pulmonary complications, and treatment with (neo)adjuvant chemoradiotherapy [4].

These findings suggest a strong association between parenchymal pathology (pre-existing lung disease, smoking history, and impaired pulmonary function tests) and the development of CPA after lung cancer surgery. Associations between intra-operative factors and post-operative complications remain tentative and less well established. In this cohort, most patients developed CPA on the same side as the operation, suggesting that surgery and possible complications are the most important determinants of subsequent CPA. Pathogenic mechanisms may include structural changes following parenchymal resection, dead space formation, prolonged air leak, and bronchopleural fistulae, with impaired immunity and poor nutritional status also playing an important role [4]. However, a small number had CPA diagnosed on the contralateral side, highlighting the potential importance of non-operative factors such as underlying parenchymal lung disease. Overall, all existing studies (including this one) are limited by patient numbers and the small number of events in the CPA groups. Thus, robust multivariable analysis with a sufficient range of clinically relevant variables is precluded.

Several limitations were identified in the current study. Firstly, our data did not capture the cause of death for patients due to the logistical difficulties in capturing this data in a cohort of patients living across the country and treated through a national centre like the National Aspergillosis Centre. Therefore, it was not possible to ascertain whether deaths in the CPA group were due to CPA rather than other causes such as underlying lung malignancy. However, both groups were well matched to reduce the impact of this on our results. Some patients identified in the NAC database underwent surgery in other centres and therefore they were absent from the NCORR database, which limited our cohort with meaningful survival data for analysis to 11/32 patients. Data were also lacking in re-operation rates and the details of (neo)adjuvant lung cancer treatment due to the retrospective dataset drawn from multiple sources, which could have an impact on mortality. Post-operative patients within the NCORR database were also not routinely screened for CPA, which may lead to an under-represented prevalence of CPA following lung resection for primary NSCLC. Additionally, patients within the local NCORR database may have been diagnosed with CPA in a centre other than the National Aspergillus Centre, which would also contribute to an under-representation of CPA prevalence following lung cancer surgery.

## 5. Conclusions

CPA is an uncommon complication following lung cancer surgery which may have an impact on survival. It can develop years after the surgery and may develop sooner in the patients who have had adjuvant chemotherapy. It can occasionally affect the contralateral side. Underlying parenchymal lung disease is an important risk factor, which may explain the high mortality rates compared with the patients with other forms of CPA. Prior squamous cell carcinoma was linked with a worse survival following CPA diagnosis. There is emerging evidence to suggest that the development of CPA after lung resection may have an adverse impact on long-term outcomes, although high-quality evidence is lacking. Further large-scale national registry data with sufficient patient numbers are required to conduct additional work on this important topic.

## Figures and Tables

**Figure 1 jof-10-00335-f001:**
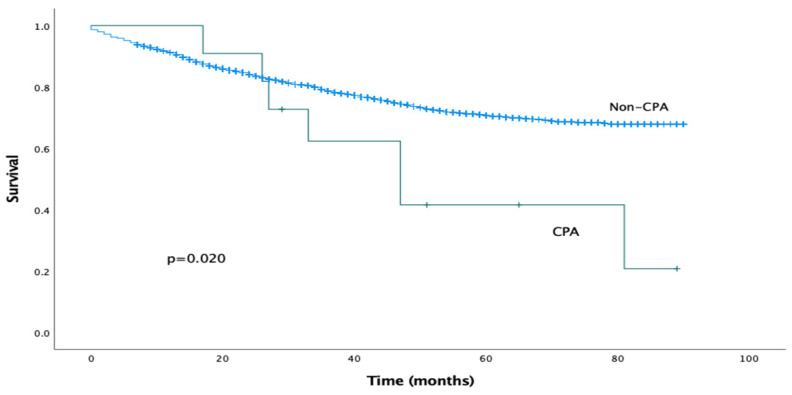
Kaplan–Meier survival curves comparing overall survival for CPA and non-CPA patients.

**Table 1 jof-10-00335-t001:** Patient characteristics.

Variable	Non-CPA (N = 4414)	CPA (N = 11)	*p* Value	Missing Data
Age (years) (mean ± SD ^a^)	66.7 (±10.8)	67.3 (± 6.5)	0.869	0%
Male sex	48.3% (n = 2130)	45.5% (n = 5)	0.853	0%
History of cancer	41.0% (n = 1809)	9.1% (n = 1)	0.032	3.90%
ASA ^b^ score (median ± IQR ^c^)	3.0 (2.0–3.0)	3.0 (2.0–3.0)	0.959	1.20%
PS ^d^ score (median ± IQR)	1.0 (0.0–1.0)	1.0 (0.0–1.0)	0.637	2.30%
NYHA ^e^ score (median ± IQR)	1.0 (0.0–2.0)	1.0 (0.0–1.0)	0.621	2.50%
% Predicted FEV1 ^f^ (mean ± SD)	88.3% (±20.9%)	71.8% (±14.9%)	0.009	10.70%
% Predicted FVC ^g^ (mean ± SD)	104.1% (±19.1%)	107.6% (±21.6%)	0.537	13.40%
% Predicted DLCO ^h^ (mean ± SD)	74.4% (±17.0%)	64.7% (±16.9%)	0.062	18.40%
BMI ^i^ (mean ± SD)	26.9 (±5.0)	25.3 (±4.5)	0.31	11.70%
Creatinine (median ± IQR)	72.0 (64.0–83.1)	73.0 (59.0–88.0)	0.913	11.60%
Anaemia	22.7% (n = 1004)	9.1% (n = 1)	0.28	12.00%
Diabetes mellitus	12.9% (n = 571)	0% (n = 0)	0.201	2.50%
Hypercholesterolaemia	16.0% (n = 707)	27.3% (n = 3)	0.31	2.50%
Hypertension	35.0% (n = 1546)	54.5% (n = 6)	0.175	2.50%
Smoking	73.2% (n = 3232)	81.8% (n = 9)	0.52	2.50%
Arrhythmia	5.9% (n = 260)	9.1% (n = 1)	0.653	4.80%
Ischaemic heart disease	13.0% (n = 573)	36.4% (n = 4)	0.021	4.80%
COPD ^j^	27.5% (n = 1214)	63.6% (n = 7)	0.007	2.80%
Cerebrovascular disease	6.5% (n = 285)	0% (n = 0)	0.384	6.40%
Peripheral vascular disease	5.2% (n = 230)	27.3% (n = 3)	0.001	1.10%
Right-sided resection	60.3% (n = 2662)	63.6% (n = 7)	0.822	0%
Complex lobectomy	7.2% (n = 319)	9.1% (n = 1)	0.812	0%
Pneumonectomy	4.4% (n = 193)	0% (n = 0)	0.478	0%
Resected segments (mean ± SD)	3.6 (±2.0)	4.2 (±1.0)	0.331	0%
Stage I/II disease	80.2% (n = 2739)	72.7% (n = 8)	0.537	0%
Any nodal disease	27.4% (n = 937)	27.3% (n = 11)	0.991	0%

^a^: standard deviation, ^b^: American Society of Anaesthesiologists, ^c^: interquartile range, ^d^: performance status, ^e^: New York Heart Association, ^f^: forced expiratory volume in 1 s, ^g^: forced vital capacity, ^h^: the diffusion capacity of the lung for carbon monoxide, ^i^: body mass index, ^j^: chronic obstructive pulmonary disease.

**Table 2 jof-10-00335-t002:** Univariable analysis of risk factors associated with CPA development.

Variable	Odds Ratio	95% Confidence Intervals		*p* Value
Age	1.005	0.949	1.063	0.869
Male sex	0.894	0.272	2.932	0.853
% Predicted FEV1 ^a^	0.96	0.931	0.99	0.01
% Predicted DLCO ^b^	0.965	0.929	1.002	0.062
BMI ^c^	0.934	0.819	1.065	0.307
Creatinine	1.002	0.983	1.023	0.81
Anaemia	0.34	0.043	2.656	0.303
Hypertension	2.226	0.678	7.306	0.187
Smoking	1.646	0.355	7.628	0.524
IHD ^d^	3.83	1.118	13.126	0.033
COPD ^e^	4.613	1.348	15.786	0.015
Right-sided resection	1.152	0.337	3.94	0.822
Resected segments	1.142	0.874	1.492	0.331
Thoracotomy	1.138	0.245	5.277	0.869
Extended resection	0.762	0.097	5.966	0.796
Stage III disease	1.515	0.401	5.726	0.54
Post-op LRTI ^f^	0.86	0.11	6.73	0.885
Post-op AF ^g^	3.58	0.769	16.653	0.104

^a^: expiratory volume in 1 s, ^b^: the diffusion capacity of the lung for carbon monoxide, ^c^: body mass index, ^d^: ischaemic heart disease, ^e^: chronic obstructive pulmonary disease, ^f^: lower respiratory tract infection, ^g^: atrial fibrillation.

## Data Availability

The raw data supporting the conclusions of this article will be made available by the authors upon request.

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
