# Peer review of "Chronic Pulmonary Aspergillosis after Surgical Treatment for Non-Small Cell Lung Cancer—An Analysis of Risk Factors and Clinical Outcomes"

_jof, 2024, doi:10.3390/jof10050335_

Round 1

Reviewer 1 Report

Comments and Suggestions for Authors

The authors describe a series of CPA following surgery for NSCLC at National Aspergillosis Centre (2009-2020) in Manchester (32 patients) and, in a complementary manner, they use the data of Northwest Clinical Outcomes Research Registry (2012-2019) (4414 patients) including 11 of the 32 patients analyzed in the local series (total: 4425). Although in the manuscript the two series are presented separately, at least in the abstract there is some confusion.

The study is interesting because it discusses an uncommon entity such as post-surgical aspergillosis in oncologic patients, its risk factors and prognosis.

The local series reports a mortality of 25% and 59.4% at 1 year and 5 years, respectively, but the registry does not clearly express mortality, except for what is observed in the Kaplan-Mier curve, and why only 11 of the 32 patients in the local series were included in the registry.

For the diagnosis of CPA are included: 1) progressive cavitary changes on a chest CT scan; 2) a positive sputum culture or positive Aspergillus IgG; and 3) exclusion of alternative diagnoses. The diagnosis of chronic aspergillosis is controversial. The results show that 46.9% of the patients (15/32 of the local series) were diagnosed with aspergilloma. The "ordinary" diagnosis of aspergilloma does not include progressive cavitary changes but stable fungus ball. On the other hand, it is commented that 100% of the patients received antifungal treatment (azoles). This is not the usual recommendation for the management of patients with simple aspergilloma. How many patients were re-operated after the diagnosis of CPA?

Finally, for the etiological diagnosis purpose a positive sputum culture or positive Aspergillus IgG were used, but for the differentiation between aspergilloma and chronic cavitated aspergillosis other measures, such as galactomannan in LAB, were used.

Author Response

The authors describe a series of CPA following surgery for NSCLC at National Aspergillosis Centre (2009-2020) in Manchester (32 patients) and, in a complementary manner, they use the data of Northwest Clinical Outcomes Research Registry (2012-2019) (4414 patients) including 11 of the 32 patients analyzed in the local series (total: 4425). Although in the manuscript the two series are presented separately, at least in the abstract there is some confusion.

Many thanks for raising this point, the abstract has been amended to clarify that the databases are separate entities with some patients present in both databases.

The study is interesting because it discusses an uncommon entity such as post-surgical aspergillosis in oncologic patients, its risk factors and prognosis.

Thank you for the comment.

The local series reports a mortality of 25% and 59.4% at 1 year and 5 years, respectively, but the registry does not clearly express mortality, except for what is observed in the Kaplan-Mier curve, and why only 11 of the 32 patients in the local series were included in the registry.

Mortality rates have been added for the registry. Additionally, a sentence has been added to the limitations to explain only 11/32 patients were included in the registry because they had surgery at other centres and therefore survival data was not available for them.

For the diagnosis of CPA are included: 1) progressive cavitary changes on a chest CT scan; 2) a positive sputum culture or positive Aspergillus IgG; and 3) exclusion of alternative diagnoses. The diagnosis of chronic aspergillosis is controversial. The results show that 46.9% of the patients (15/32 of the local series) were diagnosed with aspergilloma. The "ordinary" diagnosis of aspergilloma does not include progressive cavitary changes but stable fungus ball. On the other hand, it is commented that 100% of the patients received antifungal treatment (azoles). This is not the usual recommendation for the management of patients with simple aspergilloma. How many patients were re-operated after the diagnosis of CPA?

Many thanks for your comments. Patients were diagnosed with CPA using international clinical guidelines as per reference #7 – the manuscript has been amended to state that international guidelines were used. Furthermore, use of oral azole treatment is guided by our local guidelines – this has also been amended in the manuscript. Unfortunately, data on re-operation rates were missing – this has been added to the limitations.

Finally, for the etiological diagnosis purpose a positive sputum culture or positive Aspergillus IgG were used, but for the differentiation between aspergilloma and chronic cavitated aspergillosis other measures, such as galactomannan in LAB, were used.

All patients with new radiological findings had CT scans reviewed by a chest radiologist and cases were discussed at local respiratory and lung cancer multidisciplinary team meetings – this is stated in the methods section and helped differentiate between aspergilloma and chronic cavitated aspergillosis.

Reviewer 2 Report

Comments and Suggestions for Authors

The manuscript jof-2955611 presents an interesting perspective on the development of chronic pulmonary aspergillosis as complication of lung cancer surgery. Few comments to add:

Lines 63-66: The authors refer that “CPA was diagnosed for patients with progressive cavitary changes on a chest CT scan; a positive sputum culture or positive Aspergillus IgG and exclusion of alternative diagnoses. All the three criteria must be fulfilled for CPA diagnosis?

Line 100: A total of 32 patients were diagnosed with CPA from how many patients in total?

Line 106: “…and 74.2% (n=23/31) underwent an upper lobe resection.” Was it 31 or 32?

Line 142: “From the 32 patients described above, 11 were included in the NCORR database…” Why were those included?

Table 1: Please add (N=11) under CPA

Discussion section: References should appear in brackets

Line 199: “However, a proportion of patients died before assessment of response to CPA could be undertaken”. How many patients died? %? Are these numbers in accordance with the literature?

Line 226: “…suggesting that surgery and possible complications are the most important determinant of subsequent CPA.” Which factors may be the cause of the development of CPA following surgery? Iatrogenic factors? Environmental? Local immunological changes?

Author Response

The manuscript jof-2955611 presents an interesting perspective on the development of chronic pulmonary aspergillosis as complication of lung cancer surgery. Few comments to add:

Lines 63-66: The authors refer that “CPA was diagnosed for patients with progressive cavitary changes on a chest CT scan; a positive sputum culture or positive Aspergillus IgG and exclusion of alternative diagnoses. All the three criteria must be fulfilled for CPA diagnosis?

This has been clarified in the manuscript to confirm that all three diagnostic criteria must be met to obtain a CPA diagnosis.

Line 100: A total of 32 patients were diagnosed with CPA from how many patients in total?

Unfortunately, this data is not available.

Line 106: “…and 74.2% (n=23/31) underwent an upper lobe resection.” Was it 31 or 32?

Thank you for identifying this error, it has been corrected in the manuscript.

Line 142: “From the 32 patients described above, 11 were included in the NCORR database…” Why were those included?

This has been clarified to state that these patients were cross-referenced between databases and 11/32 were also present in the NCORR database.

Table 1: Please add (N=11) under CPA

Thank you for identifying this error, it has been corrected in the manuscript.

Discussion section: References should appear in brackets

Thank you for identifying this error, it has been corrected in the manuscript.

Line 199: “However, a proportion of patients died before assessment of response to CPA could be undertaken”. How many patients died? %? Are these numbers in accordance with the literature?

Four (12.5%) patients died before assessment of response to CPA treatment as stated in section 3.3 Outcomes. This has been added to the sentence above in the manuscript. Due to lack of data we were unable to establish the time period between starting treatment and death, making this specific comparison with the literature (which typically uses set time periods such as 1 year mortality) difficult.

Line 226: “…suggesting that surgery and possible complications are the most important determinant of subsequent CPA.” Which factors may be the cause of the development of CPA following surgery? Iatrogenic factors? Environmental? Local immunological changes?

Many thanks for raising this point. The following text has been added to the discussion: “Pathogenic mechanisms may include structural changes following parenchymal resection, dead space formation, prolonged air leak, and bronchopleural fistulae, with impaired immunity and poor nutritional status also playing an important role.”